

# Conditions for fully gapped topological superconductivity in topological insulator nanowires

**Fernando de Juan[1,2,3⋆], Jens H. Bardarson[4] and Roni Ilan[5]**

**1** Rudolf Peierls Centre for Theoretical Physics,
Oxford University, Oxford, OX1 3PU, United Kingdom
**2** Donostia International Physics Center,
P. Manuel de Lardizabal 4, 20018 Donostia-San Sebastian, Spain
**3** IKERBASQUE, Basque Foundation for Science,
Maria Diaz de Haro 3, 48013 Bilbao, Spain
**4** Department of Physics, KTH Royal Institute of Technology,
Stockholm 10691, Sweden
**5** Raymond and Beverly Sackler School of Physics and Astronomy,
Tel-Aviv University, Tel-Aviv 69978, Israel

⋆ fernando.dejuan@dipc.org

## Abstract

Among the different platforms to engineer Majorana fermions in one-dimensional topological superconductors, topological insulator nanowires remain a promising option. Threading an odd number of flux quanta through these wires induces an odd number of surface channels, which can then be gapped with proximity induced pairing. Because of the flux and depending on energetics, the phase of this surface pairing may or may not wind around the wire in the form of a vortex. Here we show that for wires with discrete rotational symmetry, this vortex is necessary to produce a fully gapped topological superconductor with localized Majorana end states. Without a vortex the proximitized wire remains gapless, and it is only if the symmetry is broken by disorder that a gap develops, which is much smaller than the one obtained with a vortex. These results are explained with the help of a continuum model and validated numerically with a tight binding model, and highlight the benefit of a vortex for reliable use of Majorana fermions in this platform.



# 1  Introduction

The physical realization and manipulation of non-Abelian anyons, exotic quasiparticles with neither fermionic nor bosonic statistics, has remained a challenging endeavor in condensed matter physics for decades [1]. Their search continues motivated both by the fundamental aim of discovering new phases of matter and by promising applications in topological quantum computation. The simplest of these quasiparticles, localized Majorana bound states, can appear in defects or on boundaries of topological superconductors [2–5]. These systems are however rare because they require unconventional pairing, but the more recent realization that they can be engineered artificially from more standard components has triggered a renewed effort to find them.

Currently, the most developed proposals are based on one dimensional (1D) superconductors which host Majorana bound states at their ends [6–8], engineered by coupling a metallic 1D system with an odd number of channels at the Fermi level with a superconductor via the proximity effect [9]. The realization of this proposal with semiconductor wires has provided compelling experimental evidence of Majorana bound states (see Ref. [10] and references therein), but several alternative realizations remain promising as well [11–13] .

One particularly interesting system that remains relatively unexplored is based on three dimensional topological insulator (TI) nanowires. When the Fermi level lies in the bulk gap, the only transport modes are those derived from the topological surface Dirac fermion [14,15] which wraps around the surface of the wire. When a flux of $h/2e$ (half of the Aharonov-Bohm flux quantum $h/e$) is threaded through the wire cross section, there is an odd number of modes at the Fermi level for any value of the chemical potential, and one of them is perfectly transmitted in the presence of time-reversal symmetry [16–22]. These wires were proposed to realize a topological superconductor in the presence of proximity effect [23,24] and this proposal has been since studied extensively [25–28]. In particular, the advantages of TI nanowires to build a Majorana qubit architecture for quantum computation were emphasized in a recent proposal [29]. Experimentally, TI nanowires have been realized in several compounds [30–38], where Aharonov-Bohm oscillations of the conductivity reveal that good surface transport has been achieved. The recent observation of Andreev reflection from the surface modes in a nanowire Josephson junction made with TI $BiSbSeTe_2$ [39,40] further supports the idea that topological superconductivity in this system should be within reach.

A key aspect of the proximity effect in wires is that the induced pairing field may acquire an azimuthal phase dependence when the magnetic field is applied, as illustrated in Fig. 1. If the intrinsic superconductor surrounds the wire as in Fig. 1(a), an azimuthal supercurrent

will develop upon flux threading, with a tendency to screen the applied flux. As the flux increases, it will become energetically favorable to switch to a state with an azimuthal vortex in the order parameter and no supercurrent, which should be most stable for an applied flux of $h/2e$. The proximity induced pairing will naturally inherit this phase profile. However, if the intrinsic superconductor is a thin film contacting one face of the wire as in Fig. 1(b), in the simplest approximation the phase profile of the order parameter and induced pairing field may be assumed constant at any flux. In a realistic setup, the vortex may or may not be present depending on flux, the device, and on material details, and the impact of the vortex on the resulting proximitized state is not sufficiently understood.

As emphasized in Ref. [26], the low energy surface Dirac model necessarily predicts that the vortex is required to produce a topological superconductor. This is because at any finite flux, the lowest energy electron mode has angular momentum of $l = 1/2$ and simply cannot be gapped out with its hole partner of $l = -1/2$ if the pairing field is constant and angular momentum is conserved. The vortex is required to compensate the mismatch in angular momentum and open a gap. In the absence of a vortex, the spectrum remains gapless and localized Majorana bound states cannot be defined. This conclusion is at odds with bulk tight binding simulations, which predict that a gapped state can be achieved without a vortex [24]. Another work with a more realistic account of the proximity effect observed both gapped and gapless regions in the absence of the vortex [27]. A better understanding of this problem is thus clearly needed.

In this work, we show with both a low energy model and tight binding calculations that in the presence of any rotation axis $C_n$ parallel to the field, which requires angular momentum conservation modulo $n$, the superconducting state without a vortex must indeed be either trivial or gapless, regardless of the proximity induced pairing strength. We then show how breaking the rotation symmetry may still lead to a gapped topological state in the absence of a vortex. However, the gap magnitude in this case is determined by the symmetry breaking mechanism and is generally much smaller than the magnitude of the pairing strength. In the presence of a vortex, on the contrary, the gap remains of the order of the pairing strength, so the topological superconducting phase is in practice much more robust in this case. We illustrate these points in detail by computing phase diagrams of the gaps and topological phase transitions for several types of wires and pairing potentials, also taking into account the effect of disorder.

The rest of this work is organized as follows. In Sec. 2 we describe the surface effective model from which all our main conclusions can be derived. In Sec. 3 we confirm these results with a lattice tight binding model with proximity effect, considering a number of scenarios. Finally in Sec. 4 we discuss our results and present some conclusions. Several technical derivations are left for the Appendix.

## 2   Continuum model for TI nanowire surface states

Topological insulators are guaranteed to possess a Dirac fermion surface state [14] which dominates their transport properties when the chemical potential $\mu$ is in the bulk band gap $E_g$. This surface state decays into the bulk within a length scale $v_F/E_g$ which is of the order of a several nm, so for bulk insulating wires of sizes much larger than this length, a surface model is enough to account for all transport properties. This model takes the form of a Dirac Hamiltonian in the corresponding surface geometry [16–18, 24, 26]. To discuss superconductivity in wires, we consider a cylindrical sample of radius $R$ and cross section $A = \pi R^2$ oriented along the $x$ direction. The surface of the wire is parametrized in cylindrical coordinates $(x, \theta)$. The wire is in the presence of a magnetic field $\vec{B} = (B_\parallel, 0, 0)$ which threads a dimensionless flux

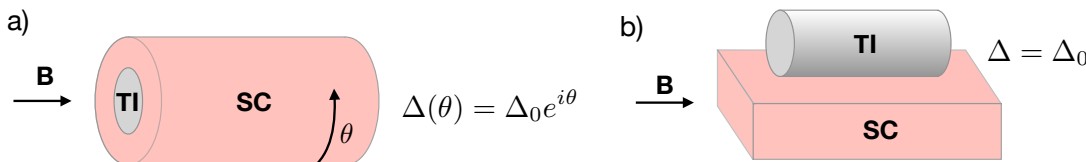

Figure 1: Two simplified setups used to induce the proximity effect in wires in the presence of a magnetic field. In a), the wire is surrounded by a superconducting cylinder, which must itself host a vortex when the flux is $h/2e$. The TI then inherits the vortex profile in proximity-induced pairing field. In b), the wire is only partially contacted by the bulk superconductor. In the limit where the superconductor is a thin film no vortices are expected, and the induced pairing in the wire will be approximately constant.

$\eta = B_\parallel A/(h/e)$ through the cross section. The magnetic field is described with the vector potential $\vec{A} = B_\parallel(0, -z/2, y/2)$, so that translational invariance is preserved in the $x$ direction. The effect of the Zeeman coupling is not essential and will be discussed in Sec. 4. The effective Dirac equation for the surface states is $H\psi = E\psi$ with Hamiltonian

$$H = -i\sigma_x \partial_x + \tfrac{1}{R}\sigma_y(-i\partial_\theta + \eta),$$ (1)

where $\sigma_i$ are Pauli matrices acting on the effective spin degree of freedom of the surface states, and we set $\hbar = 1$ and the Fermi velocity $v_F = 1$. In these units, $1/R$ is the natural energy unit for the problem. The wave functions satisfy antiperiodic boundary conditions in $\theta$ due to the curvature-induced $\pi$ Berry phase [17,18]. There are different approaches to to derive this Hamiltonian [16–18,24,26], but in all cases coordinate transformations and spin rotations can be used to bring the Hamiltonian into this form, even for a smooth cross section different from a circle. The different approaches and their relation are summarized in Appendix A.1. This model has an effective full rotational symmetry around the wire axis $\theta \to \theta + \theta'$ for any $\theta'$. The Hamiltonian can be diagonalized by Fourier transforming the spinor $\psi(x, \theta) = \int dx \sum_n e^{ikx} e^{il\theta} \psi_{k,l}$, where $l$ is half-integer, $l = \pm\frac{1}{2}, \pm\frac{3}{2}\ldots$, because of the antiperiodic boundary conditions. The $l$-th block of the transformed Hamiltonian is

$$H_l = \sigma_x k + \tfrac{1}{R}(l - \eta)\sigma_y.$$ (2)

When $\eta = 1/2$, which corresponds to half of a flux quantum threaded through the wire, the $l = 1/2$ mode is gapless, linearly dispersing, and perfectly transmitted [20], while the rest of the modes are doubly degenerate and gapped.

## 2.1 Superconductivity in the continuum model

Since $l = 1/2$ is the only non-degenerate mode at $\eta = 1/2$, the number of channels is odd for any chemical potential, and including superconducting pairing through the proximity effect should result in a topological superconductor according to Kitaev [9], as long as the resulting spectrum becomes gapped by the pairing. To see whether the spectrum becomes gapped we model superconductivity with a BdG Hamiltonian $\mathcal{H} = \frac{1}{2}\Psi^\dagger H\Psi$ written in terms of Nambu spinors $\Psi = \begin{pmatrix} \psi \\ -i\sigma_y(\psi^\dagger)^T \end{pmatrix}$ and

$$H = \left[-i\sigma_x\partial_x + \tfrac{1}{R}\sigma_y(-i\partial_\theta + \eta\,\tau_z) - \mu\right]\tau_z + \tau_x\text{Re}[\Delta(x,\theta)] + \tau_y\text{Im}[\Delta(x,\theta)],$$ (3)

where $\tau_i$ are Pauli matrices in the Nambu space (see Appendix A.2). By the BdG construction, this Hamiltonian has a particle-hole symmetry $H = -U_C H^* U_C^\dagger$, with the unitary part $U_C = \sigma_y \tau_y$. The complex pairing function $\Delta(x,\theta)$ depends on the way the proximity effect is realized. In particular, as discussed in Fig. 1, $\Delta(x,\theta)$ may have a phase winding around the perimeter of the wire. If the wire is surrounded by a superconducting shell, it is natural that this shell develops a phase winding at certain values of the flux, which is transferred to the induced pairing $\Delta(x,\theta) = \Delta_0 e^{in_v\theta}$, with $n_v$ the number of vortices. In the geometry of a wire lying on top of a flat, bulk superconductor, one may rather expect a roughly homogeneous order parameter which can be approximated by a constant $\Delta(x,\theta) = \Delta$, so $n_v = 0$.

The Hamiltonian in Eq. (3) appears to break rotation symmetry because of the $\theta$ dependence of the pairing term with generic $n_v$, but this symmetry is explicitly recovered by making the gauge transformation $\Psi \to e^{-i\tau_z n_v \theta/2}\Psi$, which results in the Hamiltonian

$$H' = \left[-i\sigma_x \partial_x + \tfrac{1}{R}\sigma_y(-i\partial_\theta + (\eta - \tfrac{n_v}{2})\,\tau_z) - \mu\right]\tau_z + \tau_x \Delta_0. \tag{4}$$

Crucially, this gauge transformation preserves the antiperiodic boundary condition for even $n_v$, while it changes it to periodic boundary conditions for odd $n_v$. The Fourier transformed Hamiltonian takes the form

$$H'_l = \left[\sigma_x k + \tfrac{1}{R}(l - (\eta - \tfrac{n_v}{2})\tau_z)\sigma_y - \mu\right]\tau_z + \tau_x \Delta_0, \tag{5}$$

where $l = \pm\tfrac{1}{2}, \pm\tfrac{3}{2}\ldots$ if $n_v$ is even, while $l = 0, \pm 1, \ldots$ if $n_v$ is odd. After the Fourier transform, particle-hole symmetry takes the form $H_{k,l} = -U_C^\dagger H_{-k,-l}^* U_C$, i.e., it reverses the angular momentum. When $n_v$ is odd, the $l = 0$ sector is special because it maps into itself under particle-hole symmetry.

This surface Hamiltonian has an inversion symmetry $H_{k,l} = U_I^\dagger H_{-k,l} U_I$ with $U_I = \sigma_y$, which will be present if the original bulk model has inversion symmetry. In addition, for the special value of the flux $\eta = n_v/2$, this Hamiltonian has an effective time-reversal symmetry $H_{k,l} = U_T^\dagger H_{-k,-l}^* U$ with $U_T = i\sigma_y$. The combination of these two symmetries when $\eta = n_v/2$ enforces that all pairs of bands with angular momentum $\pm l$ are degenerate for all $k$, except for the $l = 0$ band when it is present.

The critical role of the vortex in this problem is best illustrated with the simplest example considered in Ref. [26] when only one mode is occupied. When $\Delta_0 = 0$, the spectra of the Hamiltonian in Eq. (5) do not depend on $n_v$, while the angular momentum labels $l$ of each band do. An example spectra for $\eta = 1/2$, for a chemical potential where only the first mode is occupied is shown in Fig. 2(a). When pairing is included, however, the spectra are markedly different for $n_v = 0$ and $n_v = 1$. For $n_v = 0$ the spectrum remains gapless, because the electron branch at the Fermi level has $l = 1/2$, while the hole branch has the different angular momentum $l = -1/2$, and different angular momentum sub-blocks in the Hamiltonian cannot be mixed by the constant pairing, which preserves rotation symmetry. This is shown in Fig. 2(b). For $n_v = 1$, however, the electron and hole branches at the Fermi level are particle-hole conjugates with $l = 0$, and the pairing can gap them out, as shown in Fig. 2(c). In essence, the vortex has provided the extra unit of angular momentum to compensate the mismatch in the absence of the vortex.

The general case for arbitrary $l$, $\eta$, and $n_v$ is as follows. The energies depend on these parameters only through the combination $(l - (\eta - n_v/2))^2$. Consider first the case $\eta = n_v/2$ with effective time-reversal symmetry. If $n_v$ is even, $l$ is half integer and all electron states come in degenerate pairs of angular momentum $\pm l$. When adding superconductivity, the hole states have angular momentum $\mp l$, and because of the exact degeneracy imposed by time reversal and inversion symmetries, electron and hole states of angular momentum $l$ cross exactly at the Fermi level, and so do electron and hole states of $-l$. Therefore, infinitesimal pairing is able to

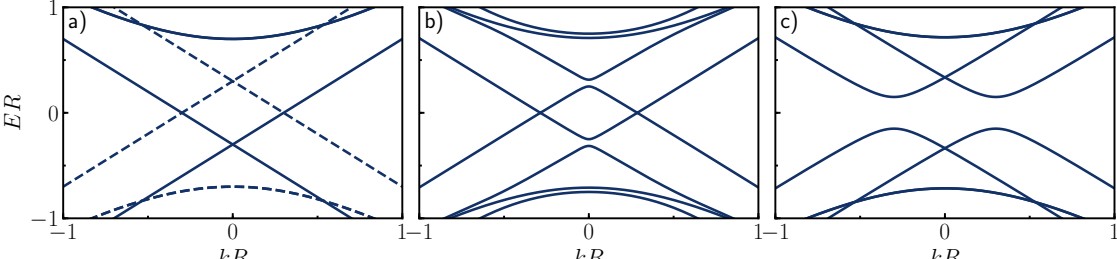

Figure 2: Spectra of topological insulator nanowires obtained from the effective Hamiltonians in Eq. (5) with $\mu R = 0.3$, when a flux of $\eta = 1/2$ is applied. a) In the absence of pairing, $\Delta_0 R = 0$, the spectrum is gapless with an odd number of modes at any $E$. Electron and hole modes are shown with full and dashed lines, respectively. b) In the presence of a paring field with $n_v = 0$, the spectrum remains gapless due to angular momentum conservation. c) In the presence of a vortex $n_v = 1$, the spectrum becomes gapped. In b) and c) $\Delta_0 R = 0.15$.

gap out both pairs. The resulting state is always gapped and trivial. If $n_v$ is odd, however, $l$ is an integer and now the non-degenerate mode $l = 0$ is allowed while the rest of integers come in degenerate pairs $\pm l$. The mentioned degeneracy allows every $|l| \leq 1$ mode to be gapped out (and $l = 0$ can always gap out as its hole partner has also $l = 0$) and since the total number of modes is always odd, the resulting state is always topological and gapped for *any* chemical potential.

As we move away from $\eta = n_v/2$, the $\pm l$ states split in energy and the ability to gap them out with their corresponding holes depends on the strength of the pairing. For sufficiently large $\eta - n_v/2$, a transition to a gapless state always occurs. The first conclusion of the effective model is thus clear: an odd number of vortices is needed for superconductivity to gap out the system for half-integer flux $\eta$. In particular, note that this means that if $n_v = 0$, it is impossible to get a topological state in the presence of rotation symmetry.

## 2.2 Computation of topological invariant

In the previous section we used Kitaev's weak-coupling mode-counting argument to decide when the system was in a topological phase. We now determine this explicitly by computing the Pfaffian topological invariant $\nu$, sometimes called the Kitaev or Majorana number [9]. This invariant is formally defined only for lattice systems with a full gap throughout the Brillouin Zone, and can be computed from the Hamiltonian matrix in the following way. First, a unitary transformation is used to express the Hamiltonian in the basis where particle-hole symmetry operation takes the simple form $H(k) = -H^*(-k)$, known as the Majorana basis. In this basis, $H$ is purely imaginary and antisymmetric at the particle-hole invariant momenta $k = 0$ and $k = \pi$. The invariant is then computed as the product of Pfaffians

$$\nu = \text{sign} \left[ \text{Pf}[iH(0)] \text{Pf}[iH(\pi)] \right]. \tag{6}$$

This invariant can only change when there is a gap closing at either $k = 0$ or $k = \pi$.

For lattice Hamiltonians modeling bulk 1D systems, this invariant can only be non-trivial when time-reversal symmetry is broken (formally in Hamiltonians of class D [4]). Since time-reversal invariance enforces Kramers degeneneracies at $k = 0, \pi$ in a 1D lattice system, the bands must connect these degeneracies such that there are always an even number of Fermi points between $k = 0$ and $k = \pi$ and superconductivity is trivial. The only way to have an odd

number of Fermi points with time reversal symmetry in a 1D system is when this is not a bulk 1D system but the 1D boundary of a higher dimensional lattice, as in the well known example of the helical edge state of a 2D topological insulator [41]. In this case, a time-reversal invariant 1D continuum Hamiltonian can be found with a single Fermi point, and superconductivity in this system is indeed topological and features Majorana edge modes.

The continuum model for the surface states of a TI nanowire assumes that any bands at $k = \pi$ are far away in energy and are never involved in superconductivity, and therefore changes in the topological invariant can be tracked by computing the Pfaffian at $k = 0$. One may thus wonder how the Pfaffian can be non-trivial in the presence of time-reversal symmetry when $\eta = n_v/2$. The reason is the same as in the case of the 2D TI edge: the 1D system under consideration is not a bulk 1D lattice, but the edge of a higher dimensional system, and it is allowed to have a single Fermi point.

We now proceed to compute the Pfaffian invariant. For any particle-hole invariant block diagonal Hamiltonian, the Pfaffian can be decomposed as the product of the Pfaffians for each block. We consider first the case with $n_v = 0$. The Hamiltonian in Eq. (5) is block diagonal in angular momentum $l$, but particle hole symmetry maps blocks with angular momentum $\pm l$ into each other, so the smallest block to compute the Pfaffian must include both $\pm l$ subblocks. Defining Pauli matrices $\alpha_i$ that act on this degree of freedom, a doubled Hamiltonian for a given $|l|$ can be written as

$$H_{|l|} = \left[ \sigma_x k + \tfrac{1}{R}(l\alpha_z - \eta \tau_z)\sigma_y - \mu \right] \tau_z + \tau_x \Delta_0 \tag{7}$$

and particle-hole symmetry is implemented as $H_{|l|}(k) = -U_C H^*_{|l|}(-k) U_C^\dagger$ with $U_c = \sigma_y \tau_y \alpha_x$. To switch to the Majorana basis, we employ a unitary transformation $H^M = U_M H U_M^\dagger$ constructed such that $U_M U_C U_M^T = 1$, so that particle-hole symmetry becomes $H^M_{|l|}(k) = -H^{M*}_{|l|}(-k)$ as required. This matrix is $U_M = U \otimes U'$ where $U$ acts on spin and particle-hole indices and is given by

$$U = \frac{1}{\sqrt{2}} \begin{pmatrix} \mathcal{I} & -i\sigma_y \\ -i\mathcal{I} & \sigma_y \end{pmatrix}, \tag{8}$$

with $\mathcal{I}$ the identity matrix and

$$U' = 1/2[(1+i)\mathcal{I} + (1-i)\alpha_x]. \tag{9}$$

In this basis, the Hamiltonian is

$$H^M_{|l|} = \sigma_x k - \tfrac{1}{R}l\alpha_y \sigma_y \tau_y - \eta \sigma_y + \mu \tau_y + \sigma_y \tau_x \Delta_0. \tag{10}$$

The Pfaffian at $k = 0$ is

$$\mathrm{Pf}[iH^M_{|l|}(0)] = [\eta^2 - R^2(\Delta_0^2 + \mu^2)]^2/R^4 + [l^4 - 2l^2(\eta^2 + R^2(-\Delta_0^2 + \mu^2))]/R^4, \tag{11}$$

and the Majorana number is

$$\nu = \mathrm{sign}\left[ \prod_{l=1/2,3/2,\ldots} \mathrm{Pf}[iH^M_{|l|}(0)] \right]. \tag{12}$$

In the case where $n_v = 1/2$ is odd, $l$ is an integer and the $l = 0$ block has to be considered separately because it transforms into itself under particle-hole symmetry. For $l \neq 0$ the Hamiltonian and the Pfaffian are the same as before except $\eta \to \eta - \tfrac{1}{2}$ and $l$ is an integer so

$$H_{|l|\neq 0} = \left[ \sigma_x k + \tfrac{1}{R}(l\alpha_z - (\eta - \tfrac{1}{2})\tau_z)\sigma_y - \mu \right] \tau_z + \tau_x \Delta_0 \tag{13}$$

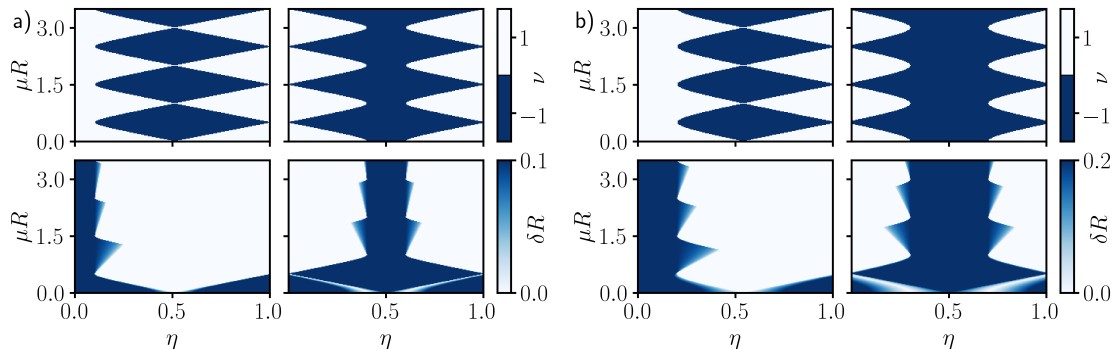

Figure 3: (a) Topological invariant (upper subpanels) and gap estimate $\delta R$ (lower subpanels) obtained from the continuum model, with a vortex absent (left subpanels) or present (right subpanels). The pairing potential is $\Delta_0 R = 0.1$. (b) is the same as a) but with $\Delta_0 R = 0.2$.

and

$$\text{Pf}[iH^M_{|l|\neq 0}(0)] = [(\eta - 1/2)^2 - R^2(\Delta_0^2 + \mu^2)]^2/R^4$$
$$+ [l^4 - 2l^2((\eta - 1/2)^2 + R^2(-\Delta_0^2 + \mu^2))]/R^4. \quad (14)$$

The Hamiltonian for $l = 0$ is transformed with Eq. (8) alone and gives

$$H^M_0 = \sigma_x k - (\eta - 1/2)\sigma_y + \mu\tau_y + \sigma_y \tau_x \Delta_0, \quad (15)$$

and the Pfaffian is

$$\text{Pf}[iH^M_0(k = 0)] = -\Delta_0^2 - \mu^2 + (\eta - 1/2)^2/R^2, \quad (16)$$

which is always negative if $\eta = 1/2$. The Majorana number is

$$\nu = \text{sign}\left[ \text{Pf}[iH^M_0(0)] \prod_{|l|\neq 0} \text{Pf}[iH^M_{|l|\neq 0}(0)] \right]. \quad (17)$$

## 2.3 Phase diagrams from continuum model

With the analytical expressions for the Pfaffian at $k = 0$, given in Eq. (12) (with no vortex) and Eq. (17) (with vortex) we can now map out phase diagrams as a function of different model parameters showing the topological and trivial regions. To do this, we note that at the point $\Delta_0 = \eta = 0$, the system is a non-superconducting insulator which must have trivial Majorana number. As we move through the phase diagram, the Majorana number will become non-trivial when the Pfaffian at $k = 0$ changes sign compared to that point.

To address the problem of whether the superconducting state is gapless, it is useful to display phase diagrams showing both the topological invariant and the gap $\delta$ induced by superconductivity. The numerical computation of the gap is complicated by the fact that it is not efficient to sweep over $k$ to find the minimum separation between bands above and below zero, especially so once we consider lattice models with many bands in the next section. Because of this, we consider an alternative method to estimate the gap based on the transfer matrix approach, explained in detail in Appendix A.3, which rather computes the values of the momentum $k$ for all modes at zero energy (at the Fermi level). Modes that do not cross

the Fermi level are evanescent and have a complex momentum $k = \kappa + i\delta$, and the imaginary part $\delta$ for a given mode can be taken as an estimate of its gap (note $v_F = 1$). Numerically, for a given point in the phase diagram we compute $\delta$ for all modes and take the smallest $\delta$ as an estimate of the true gap.

With this procedure, we compute phase diagrams as a function of flux $\eta$ and chemical potential $\mu$, which are displayed in Fig. 3 for two values of $\Delta$. A first main result is that the topological invariant, taken at face value, is not very different between the cases with and without vortex. This is in agreement with previous work [24]. However, a side by side comparison of the gap and topological invariant clearly shows that, in the absence of a vortex, all regions where the topological invariant is formally nontrivial are in fact gapless. It is only when the vortex is present that a region centered around $\eta = 1/2$ appears in the phase diagram which has both a nontrivial Majorana number and a finite gap. This gapped region, as well as the one centered around $\eta = 0$ for $n_v = 0$, extends to arbitrary chemical potential, as discussed in the previous section.

# 3   Topological superconductivity in a tight binding model

The results presented in the previous section are ultimately rooted in the full rotational invariance of the low-energy effective Hamiltonian. However, real lattice systems hosting a TI state might have at most a discrete $n-$fold rotation symmetry, with $n = 2, 3, 4, 6$ depending on the lattice point group. Even if the microscopic lattice has this symmetry, the actual device geometry or the presence of disorder might also break it. One might thus wonder to what extent the continuum model results apply to real systems.

The effect of a discrete $n-$fold axis is that it enforces angular momentum conservation only modulo $n$. This constraint is weaker than that induced but full rotations, but it can still be enough to enforce a gapless superconducting state. Consider the example of the previous section where only the lowest mode is occupied at $\eta = 1/2$, $n_v = 0$. Since the angular momentum mismatch between electron and hole state is 1, even a twofold axis (enforcing angular momentum conservation modulo 2) is enough to prevent the mixing of these two bands. If we thread a flux of $\eta = 3/2$, electron and hole branches at the Fermi level have angular momentum of $l = \pm 3/2$, with a mismatch of 3, and again any twofold axis prevents a gap opening. Notably, however, a threefold axis would not prevent a gap opening in this case, as angular momentum is conserved only modulo 3. The general logic is clear: for an even-fold rotation axis, the lowest energy mode at half-integer flux can never be gapped out by superconductivity without a vortex.

When all relevant symmetries are broken, superconductivity is allowed to generate a fully gapped state. There remains however the practical matter of how large the gap can be in this case. To illustrate the symmetry constraints and to study the effects of symmetry breaking quantitatively, we now consider a lattice model for a proximitized TI nanowire in several geometries. We consider the model in Ref. [23], with BdG Hamiltonian

$$
\begin{aligned}
H_k =& [\epsilon - 2t(\cos k_x + \cos k_y + \cos k_z)]\rho_x \tau_z + \lambda_z \rho_y \tau_z \sin k_z \\
&+ \lambda \rho_z \tau_z (s_y \sin k_x - s_x \sin k_y) + \tau_x \mathrm{Re}\Delta + \tau_y \mathrm{Im}\Delta - \mu \tau_z,
\end{aligned}
\tag{18}
$$

where $\rho, s, \tau$ denote Pauli matrices for orbital, physical spin and particle-hole degrees of freedom. This model has an inversion symmetry generated by $U_I = \rho_x$ and $\vec{k} \to -\vec{k}$, and a mirror symmetry generated by $U_{M_z} = \rho_x \sigma_z$ and $k_z \to -k_z$. The mirror symmetry will play the same role as inversion in the 1D geometry, and is preserved for any cross section of the wire such as a triangular one. The parameters $\epsilon, t, \lambda_z$ represent hopping amplitudes, $\lambda$ is a spin-orbit coupling strength, $\Delta$ is the pairing potential and $\mu$ is the chemical potential. We measure all

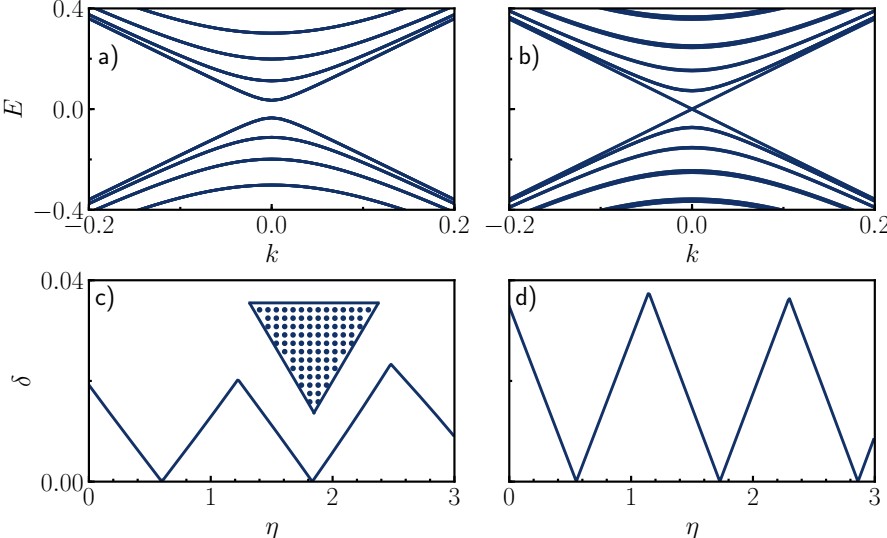

Figure 4: Spectra of the triangular topological insulator nanowire shown in the inset to c), obtained in the tight binding model with $\Delta = \mu = 0$, in the presence of magnetic flux at a) zero flux and b) $\eta = 0.6$ where it becomes gapless. The flux required for closing the gap is larger than 0.5 due to finite size effects. c) Gap as a function of flux for the same triangular wire, showing several zeros. d) Gap as a function of flux for a square wire with $N_x = N_y = 10$, showing zeros at different positions due to finite size effects.

quantities with dimensions of energy in units of the hopping $t$, and we take $\lambda = 1$, $\lambda_z = 1.8$ and $\epsilon = 4$, which realizes a topological insulator state [23]. We take a geometry where the $x$ and $y$ directions are finite, extending $N_x$ and $N_y$ sites in each direction (note in this section we take a rotated coordinate system where the wire is aligned in the $z$ direction). Denoting the site number with a discrete pair of indices $i, j$, the matrix elements of the Hamiltonian are

$$H_{(i,j),(i,j)} = [\epsilon - 2t \cos k_z]\rho_x \tau_z + \lambda_z \rho_y \tau_z \sin k_z + \tau_x \mathrm{Re}\Delta_{i,j} + \tau_y \mathrm{Im}\Delta_{i,j} - \mu \tau_z, \tag{19}$$

$$H_{(i,j),(i+1,j)} = (-t\rho_x \tau_z + i\lambda/2\rho_z s_y \tau_z)e^{i\tau_z \phi_j}, \tag{20}$$

$$H_{(i,j),(i,j+1)} = (-t\rho_x \tau_z - i\lambda/2\rho_z s_x \tau_z)e^{i\tau_z \phi_i}. \tag{21}$$

The wire is threaded by a flux described by a vector potential $\vec{A} = B_{\parallel}/2(y - y_0, -(x - x_0), 0)$, where the origin is chosen to respect the fourfold axis of the lattice. For even $N_x$ and $N_y$, the origin $(x_0, y_0)$ is chosen in the middle of the central plaquette. The phases $\phi_i$ and $\phi_j$ implement the Peierls phase for this vector potential. The pairing strength $\Delta_{i,j} = \Delta_0 e^{in_v \arctan(y-y_0)/(x-x_0)}$ is complex and may contain any number of vortices. This Hamiltonian has particle hole symmetry given by $U_c H^*(-k)U_c^\dagger = -H(k)$, with $U_c = s_y \tau_y$.

In the absence of pairing this model correctly produces a bulk insulator with a Dirac fermion surface state [24], and its lowest energy modes respond to the flux in the same way as in the effective low energy model, with the caveat that the physical value of the flux that produces a gapless spectrum might deviate somewhat from $1/2$ due to the penetration depth of the surface state into the bulk. These finite size effects are also observed in more realistic ab-initio calculations of topological insulator nanowires [42]. As an example illustrating these features, in Figs. 4(a,b) we present the spectrum of a triangular wire with shape depicted in the inset to Fig. 4(c), at $\eta = 0$ and $\eta = 0.6$ where the gap closes. The spectrum indeed reproduces that of the effective model, in particular the degeneracies with effective time-reversal symmetry. We

have chosen this triangular wire as an example with no rotation axis. The same spectrum is obtained for square wires (not shown). Figs. 4(c,d) show the minimum gap between the lowest energy bands at $k = 0$ for the triangular wire and a square wire for comparison, emphasizing that gap closings occur periodically as in the effective model, but at fluxes that depend on the wire details.

We now produce the same type of phase diagrams as for the continuum model for comparison. To compute the Kitaev number we again need to express $H$ in the Majorana basis, which is achieved by a unitary transformation $H^M = UHU^\dagger$, with $U$ now given by

$$U = \frac{1}{\sqrt{2}} \begin{pmatrix} \mathcal{I} & -is_y \\ -i\mathcal{I} & s_y \end{pmatrix}. \tag{22}$$

After this transformation the Hamiltonian is

$$\begin{aligned} H_k = & -[\epsilon - 2t(\cos k_x + \cos k_y + \cos k_z)]\rho_x\tau_y - \lambda_z\rho_y\tau_y\sin k_z - \lambda\rho_z s_y\tau_y\sin k_x \\ & -\lambda\rho_z s_x\sin k_y + s_y\tau_x\mathrm{Re}\Delta + s_y\tau_z\mathrm{Im}\Delta + \tau_y\mu \end{aligned} \tag{23}$$

and the matrix elements are

$$H_{(i,j),(i,j)} = -[\epsilon - 2t\cos k_z]\rho_x\tau_y - \lambda_z\rho_y\tau_y\sin k_z + s_y\tau_x\mathrm{Re}\Delta_{i,j} + s_y\tau_z\mathrm{Im}\Delta_{i,j} + \mu\tau_y, \tag{24}$$

$$H_{(i,j),(i+1,j)} = (t\rho_x\tau_y - i\lambda/2\rho_z s_y\tau_y)e^{-i\tau_y\phi_j}, \tag{25}$$

$$H_{(i,j),(i,j+1)} = (t\rho_x\tau_y - i\lambda/2\rho_z s_x)e^{-i\tau_y\phi_i}. \tag{26}$$

In this basis $iH$ is a real antisymmetric matrix at $k = 0, \pi$, and the Kitaev number is given by Eq. (6).

## 3.1 Phase diagrams

We now consider a number of wire geometries and setups, and present phase diagrams for these showing the topological invariant and the estimate of the gap computed from the transfer matrix as described in the previous section. Fig. 5 shows phase diagrams arranged in the same way as in Fig. 3 for the continuum model, where the left subcolumn of each panel considers pairing without a vortex, while the right subcolumn considers pairing with a vortex. In Fig. 5(a) we consider a square wire with $N_x = N_y = 10$, which has fourfold rotation symmetry. We see that the results match almost identically to the continuum model results in Fig. 3 for both subcolumns. In particular, for $n_v = 1$ we do get a gapped topological state for arbitrary values of the chemical potential, and for $n_v = 0$, the only gapped states occur around zero flux and are trivial. The presence of a fourfold axis in this case is enough to enforce gaplessness around $\eta = 1/2$ as in the continuum model, where the topological region was expected. These results are in contrast with a previous lattice calculation of essentially the same geometry [24], which did find some gapped, non-trivial regions.

Considering possible explanations for the discrepancy, in Fig. 5(b) we present the same calculation where the origin of the vector potential $(x_0, y_0)$ is displaced away from the central plaquette, keeping the phase of the pairing profile unchanged. If $\Delta_0 = 0$, this is just a choice of gauge and makes no difference in the spectrum. However, in the presence of a general pairing $\Delta(x) = \Delta_0(x)e^{i\phi(x)}$, this choice has physical consequences as the physical supercurrent is proportional to $\vec{J}_{SC} \sim \vec{A} + 2\vec{\partial}\phi$. Keeping the same $\Delta(x)$ but shifting $\vec{A}$ leads to a different supercurrent, and in particular to one that breaks the original fourfold symmetry. While we make this choice as an example, physically the supercurrent and vector potential would have to be solved for self-consistently and will depend on the applied flux. The supercurrent pattern will have more structure than our simple choice, but there is no reason to expect that it would spontaneously break the original symmetry of the problem.

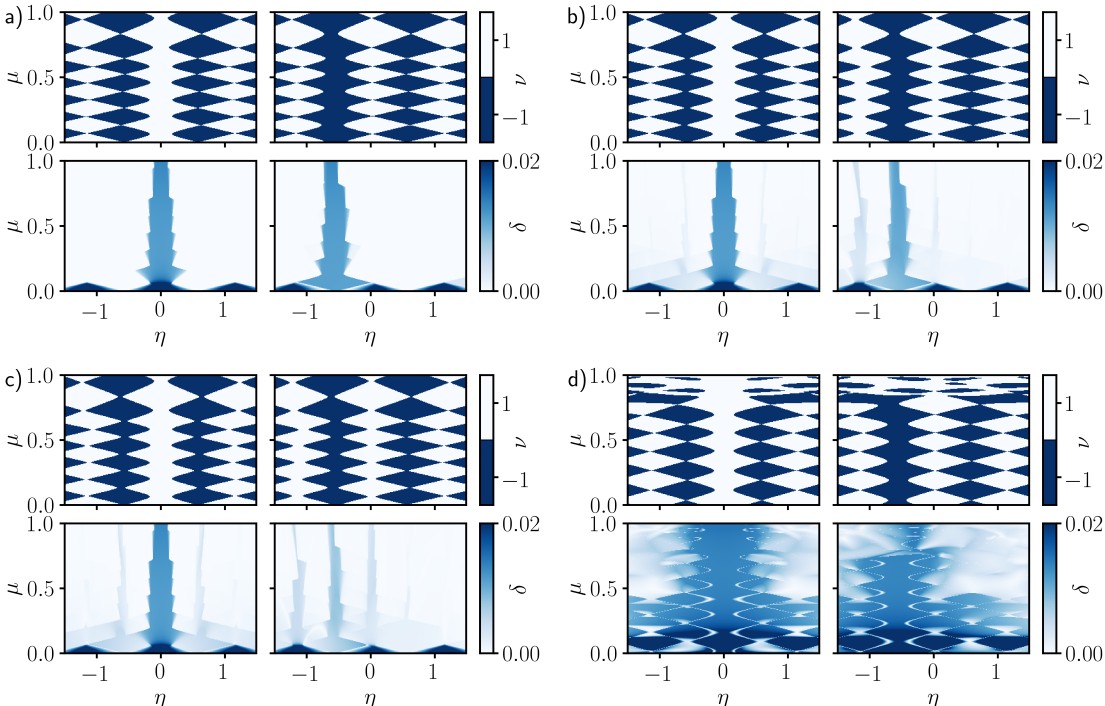

Figure 5: Phase diagram and gap obtained from the tight binding model in the presence of flux, with pairing $\Delta_0 = 0.02$. Subpanels are arranged in the same way as in Fig. 3 where analytical results are shown. a) Results for a clean, square wire with $N_x = N_y = 10$. b) Results in the presence of an asymmetric supercurrent profile, which is modeled by choosing the origin of the gauge potential offset from the center of the wire ($x_0 = 2.25$). c) Results for a disordered wire with $W = 0.1$. d) Results for a disordered wire with $W = 0.4$.

Using this vector potential, we observe that in the absence of a vortex, we now get gapped, topological regions around $\eta = 1/2$, which is only possible when the fourfold axis is broken. This reveals the importance of choosing both the vector potential and the superconducting phase in such a way that the original symmetries of the problem are respected. An oversight in this choice could be one possible explanation of the discrepancy of Ref. [24] with both the effective model and our own tight binding simulations. In practice, a supercurrent that breaks the fourfold axis can only be expected if this symmetry is already broken structurally, for example because of the presence of a substrate. A realistic simulation of the supercurrent profile induced by proximity effect in the presence of a field is beyond the scope of this paper.

We next consider disorder, a more physical mechanism that might lead to a topological superconductor without a vortex by breaking rotation symmetries. As the simplest example, we consider a wire with a potential that is constant along the wire, but which fluctuates across the wire cross section. That is, in Eq. (24) we take $\mu \rightarrow \mu + \delta\mu_{i,j}$ where $\delta\mu_{i,j}$ is a random number uniformly distributed in the range $[-W, W]$. Figs. 5(c,d) show two phase diagrams for two strengths of disorder $W$. We see that weak disorder in Fig. 5(c) again enables gapped topological regions in the absence of a vortex, but with a magnitude of the gap that is much smaller than the pairing strength. Strong disorder, shown in Fig. 5(d), allows gapped regions to emerge everywhere in the phase diagram.

We next consider further examples of the effect of symmetry breaking, now only in the absence of a vortex. Fig 6(a) shows the square wire again for reference, with an enlarged

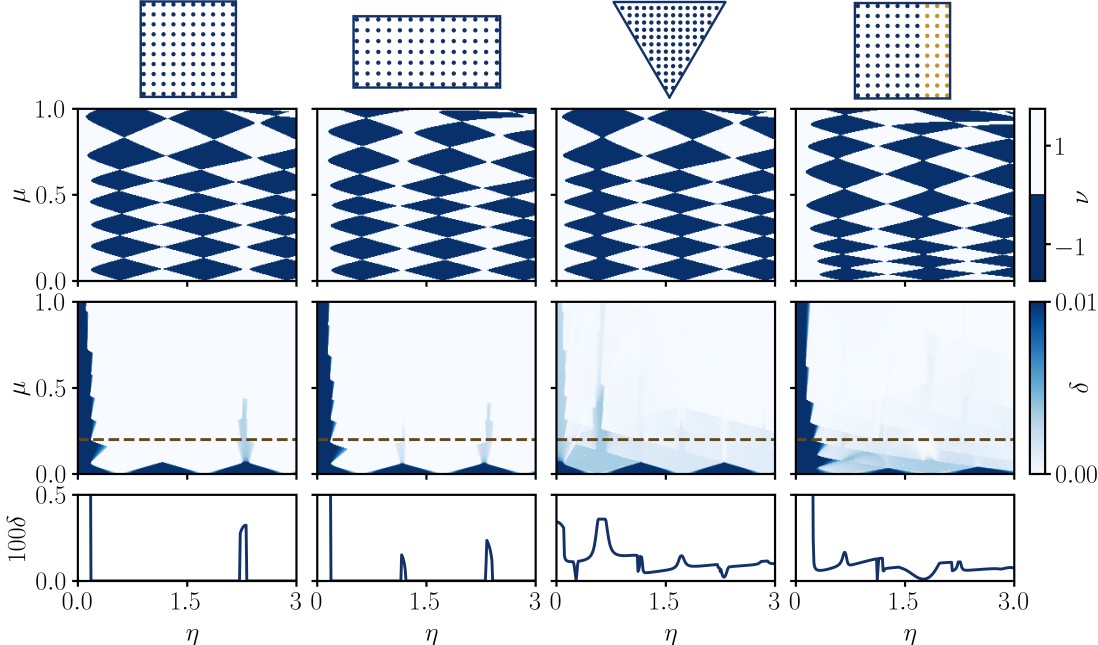

Figure 6: Topological invariant and gap for different ways of breaking the $C_4$ rotation, in the absence of a vortex in the order parameter. The top row shows the wire geometry: from left to right, the $C_4$ invariant square wire for reference; a rectangular wire with $N_x = 10$, $N_y = 7$ and only $C_2$ rotational symmetry; the triangular wire from Fig. 5, where all symmetries are lost; and a square wire where the proximity effect is finite only for sites with $N_x > 7$, representing the situation described in Fig 1. Second row is the Kitaev number, and third row the gap estimate. The fourth row zooms in on the dashed lines in the third row.

range of flux. Around an effective $\eta = 2$ certain gapped states appear, while these do not occur in the continuum model. At this flux, the electron states with angular momentum $l$ and $-l + 4$ are doubly degenerate, and the corresponding hole states have $-l$ and $l - 4$. Since now angular momentum is conserved only modulo 4, states with $l$ and $l - 4$ can pair and so can $-l$ and $-l + 4$, so gapped states are allowed. Fig. 6(b) shows a rectangular wire with only a twofold rotation symmetry. In this case, we observe another gapped region around $\eta = 1$, where the lowest two degenerate bands have an angular momentum mismatch of 2. These extra gapped regions in Figs. Fig 6(a,b) are topologically trivial. Furthermore, they only occur for a small range of chemical potentials. The reason for this is that we are rather far away from $\eta = n_v = 0$, and the effective time reversal symmetry is broken, so the degeneracies we mentioned are accidental and splittings must occur at higher chemical potentials.

Fig. 6(c) shows the same phase diagram for a triangular wire with has no rotation symmetries, where we observe that a very small gap is opened for all values of the flux. Finally, Fig. 6(d) represents a more realistic account of the situation in Fig. 1(b), where the proximity effect is only induced in the few layers closest to the bulk superconductor, and again no symmetries remain. In this case a small gap again opens for every value of the flux. The lowest row of plots show a cut of the estimated gap for a given chemical potential, emphasizing that with no rotation symmetry the gap is always finite but small.

## 4   Discussion and conclusions

The main conclusion to be drawn from this work is that a topological superconducting state can be engineered with topological insulator nanowires in magnetic fields, but the magnitude of the induced superconducting gap is strongly dependent on the device geometry, and in particular on whether there is a superconducting vortex winding around the perimeter of the wire. In the absence of such vortex, discrete rotation symmetries may enforce a gapless state, and if these symmetries are broken only weakly, the gap will be correspondingly very small. In the case where there is a vortex, however, an effective time-reversal symmetry at $\eta = n_v/2 = 1/2$ enables a fully gapped, topological region for an arbitrary value of the chemical potential. We have illustrated these point with an effective model for a TI with $C_4$ symmetry, by breaking the symmetry in different ways.

For actual devices made of the prototypical TI $Bi_2Se_3$ [15,43], with point group $D_{3d}$, similar conclusions will apply. Wires with well defined facets grown along the crystallographic $c$ axis will have a threefold symmetry if their cross section is triangular or hexagonal, while wires grown along the $a$ axis will have twofold symmetry if their cross section is rectangular [15]. This symmetries might be broken depending on the way the superconductor layer is grown. Quantitative predictions for these systems can be made with more realistic $sp^3$ tight binding models [44,45] and a more microscopic account of the proximity effect as in Ref. [27].

The effective time-reversal symmetry at $\eta = n_v/2$ can be broken in actual wires by several mechanisms, which include the Zeeman coupling and the finite extent of surface wavefunctions into the bulk which leads to orbital effects. The Zeeman coupling to the parallel field results in an extra contribution to $\eta$ [23], with the only effect that the value of the flux where the perfectly transmitted mode appears deviates from $1/2$. The Zeeman $g$ factor in these systems has been measured to be in the range 6-18 [46]. The Zeeman energy scale for the fields required to make a topological superconductor with this value is of the order of a few meV and its effect is expected to be small in any case. Orbital effects in realistic wires will also be small as the decay length of the surface modes is only a few nm for $Bi_2Se_3$.

It is also interesting to note the very different implications of effective time-reversal symmetry in our system compared with the recent proposal [47] for a topological superconductor in full-shell Rasbha-split semiconductor nanowires, recently realized experimentally [48]. In the hollow cylinder approximation, the Hamiltonian maps to the original model in Refs. [6,7], and the effect of the magnetic field comes only through the Aharonov-Bohm phase, so at $\eta = n_v/2$ there is also an effective time-reversal symmetry. However, since this is a bulk 1D system, it is a general constraint that one cannot get a class D topological superconductor in the presence of time reversal symmetry. While the effective model for those wires is apparently similar to the one used here, gapped topological regions in Ref. [47] appear only away from $\eta = n_v/2$ and tend to become gapless in the presence of several occupied modes, while in our case the region $\eta = n_v/2$ is optimal for topological superconductivity as it extends for arbitrary values of the chemical potential within the bulk gap. Time-reversal symmetry does not prevent the effective model we use from becoming a topological superconductor because the model does not represent a bulk 1D system but rather the boundary of a 3D system. Gapless superconductivity was also predicted in a related coupled wire model with threefold symmetry, which become gapped once this symmetry is broken [49].

In a transport experiment, topological superconductvity can be detected via perfect Andreev reflection in a normal-superconductor junction [26], but again it should be noted that this requires a fully gapped state. If the superconductor is gapless, quasiparticle transport contributes in addition to Andreev reflection. A fully gapped state is also required in any proposal that aims at implementing any type of braiding experiments.

In summary, in this work we have presented a detailed account of the influence of an

azimuthal vortex in the order parameter of proximitized TI nanowires. We believe that the results presented in this work can serve as a guide to a more realistic implementation of Majorana fermion networks made of topological insulators, and may stimulate further experimental developments.

# Acknowledgements

The authors would like to thank M. Franz, Y. Chen and Y. Ando for very useful discussions. F. J. was supported by the Marie Curie Programme under EC Grant agreement No. 705968. J. H. B. was supported by the ERC Starting Grant No. 679722 and the Knut and Alice Wallenberg Foundation 2013-0093. This research was supported in part by the National Science Foundation under Grant No. NSF PHY-1748958.

# A   Appendix

## A.1   Dirac equation in curved space

In this appendix we review the derivation of the effective Hamiltonian for the surface states of a TI nanowire with an arbitrary cross section, with an emphasis on unifying previous formalisms used for the problem. We consider a surface parametrized by two coordinates $y^\alpha$ with $\alpha = 1, 2$, living in three dimensional space described by coordinates $x^i$ with $i = 1, 2, 3$. Greek indices $\alpha, \beta, \cdots$ will denote surface coordinates, while latin indices $i, j, \cdots$ will denote flat space coordinates. The two basis vectors normal to the surface at every point, $\vec{e}_1$ and $\vec{e}_2$, are given by $e^i_\alpha = \partial x^i / \partial y^\alpha$. The unit normal to the surface is $\vec{n} = \vec{e}_1 \times \vec{e}_2 / |\vec{e}_1 \times \vec{e}_2|$.

A general effective Hamiltonian valid for any curved surface was first derived in the supplement of Ref. [16]. This is obtained from a 3D massive Dirac fermion model for the bulk by solving for an interface with normal $\vec{n}$ and then making $\vec{n}$ position dependent. Setting $\hbar = v_F = 1$, the Hamiltonian $\mathcal{H} = \psi^\dagger H \psi$ is given by

$$H = \frac{\vec{\nabla} \cdot \vec{n}}{2} - \frac{i}{2} \left[ \vec{n} \cdot \vec{\sigma} \times \vec{\nabla} + \vec{\sigma} \times \vec{\nabla} \cdot \vec{n} \right], \tag{27}$$

where $\vec{\nabla} = (\partial_x, \partial_y, \partial_z)$ and $\vec{\sigma} = (\sigma_x, \sigma_y, \sigma_z)$. Since $\vec{\nabla} \times \vec{n} = 0$, this can be written as

$$H = \frac{\vec{\nabla} \cdot \vec{n}}{2} - i\vec{n} \cdot \vec{\sigma} \times \vec{\nabla}. \tag{28}$$

This is also the model used in Refs. [23,24]. Since $\psi$ only depends on the surface coordinates $y^\alpha$, the gradient acts as $\nabla^i \psi = \partial y^\alpha / \partial x^i \partial_\alpha \psi = e^\alpha_i \partial_\alpha \psi$. The inverse basis vector $e^\alpha_i \partial_\alpha$ is defined with $\alpha$ as an upper index by convention. This inverse or conjugate basis satisfies $\vec{e}^\alpha \vec{e}_\beta = \delta^\alpha_\beta$. In the Hamiltonian in Eq. (28), the spin is defined with respect to the flat space, constant coordinate system given by $\hat{x}, \hat{y}, \hat{z}$ as usual. If we were to include the Zeeman coupling with respect to external field, it would take the usual form

$$H = \frac{\vec{\nabla} \cdot \vec{n}}{2} - i\vec{n} \cdot \vec{\sigma} \times \vec{\nabla} + \frac{\mu_B}{2} g \vec{\sigma} \cdot \vec{B}. \tag{29}$$

This Hamiltonian is not written in the standard form of a Dirac Hamiltonian in curved space, which was derived in Ref. [17]. To put it in this form, we can rotate the spin basis by

$\pi/2$ around the normal at each point, $\psi \to U\psi$ with $U = e^{i\frac{\vec{\sigma}\vec{n}}{2}\frac{\pi}{2}} = \frac{1}{\sqrt{2}}(1 + i\vec{\sigma}\cdot\vec{n})$. The derivative term transforms as

$$-iU^{\dagger}(\vec{n}\cdot\vec{\nabla}\times\vec{\sigma})U = -i\vec{\sigma}\cdot\vec{\nabla} + \frac{1}{2}\left(-i\vec{\sigma}\cdot\vec{n}\vec{\nabla}\cdot\vec{n} - \vec{\nabla}\cdot\vec{n}\right),$$

where we have used $2n_i\vec{\nabla}n_i = \vec{\nabla}(\vec{n}^2) = 0$ and $\vec{\nabla}\times\vec{n} = 0$. This gives the Hamiltonian

$$H = -i\left(\vec{\sigma}\cdot\vec{\nabla} + \frac{1}{2}\vec{\sigma}\cdot\vec{n}\,\vec{\nabla}\cdot\vec{n}\right). \tag{30}$$

Remembering that $\nabla_i\psi = e_i^{\mu}\partial_{\mu}\psi$ and defining curved space Dirac matrices $\alpha^{\mu} = \vec{e}^{\mu}\vec{\sigma}$, Eq. (30) takes the form of a Dirac Hamiltonian in curved space

$$H = -i\alpha^{\mu}(\partial_{\mu} + \Gamma_{\mu}), \tag{31}$$

with

$$\alpha^{\mu}\Gamma_{\mu} = \frac{1}{2}\vec{\sigma}\cdot\vec{n}\,\vec{\nabla}\cdot\vec{n}. \tag{32}$$

This form of the Dirac equation was used in Ref. [17]. There, the spin connection $\Gamma_{\mu}$ was defined in terms of the normal Pauli matrix $\beta = \vec{\sigma}\cdot\vec{n}$ as $\Gamma_{\mu} = -\frac{1}{2}\beta\partial_{\mu}\beta$. With this definition we have

$$\alpha^{\mu}\Gamma_{\mu} = \vec{\sigma}\vec{e}^{\mu}(-\frac{1}{2}\beta\partial_{\mu}\beta) = \frac{1}{2}\beta\vec{\sigma}\vec{e}^{\mu}\partial_{\mu}\beta = \frac{1}{2}\beta\vec{\sigma}\vec{\nabla}\beta = \frac{1}{2}\vec{\sigma}\cdot\vec{n}\vec{\nabla}\cdot\vec{n}, \tag{33}$$

which indeed reproduces Eq. (30). If a general Zeeman term had been included, it would have become position dependent due to the rotation $U$.

The curved space Dirac Hamiltonian is actually much simpler for a surface that has no intrinsic curvature. For our purposes, we now consider the specific surface of a straight wire, parallel to the $z$ direction and with arbitrary cross section in the x-y plane given by the function $r(\theta)$ (for a cylinder of unit radius we would take $r(\theta) = 1$). Because this surface has no intrinsic (Riemann) curvature, there is a coordinate system where this equation looks like the Dirac equation in flat space, which we now find explicitly. The basis vectors for this surface are

$$\vec{e}_1 = \hat{z}, \tag{34}$$

$$\vec{e}_2 = \frac{\partial x}{\partial\theta}\hat{x} + \frac{\partial y}{\partial\theta}\hat{y} = (r'\cos\theta - r\sin\theta)\hat{x} + (r'\sin\theta + r\cos\theta)\hat{y}, \tag{35}$$

with $r' = \partial_{\theta}r$. The conjugate (upper index) basis satisfying $\vec{e}^i\vec{e}_j = \delta_j^i$ is

$$\vec{e}^1 = \hat{z}, \tag{36}$$

$$\vec{e}^2 = \frac{r'\cos\theta - r\sin\theta}{r'^2 + r^2}\hat{x} + \frac{r'\sin\theta + r\cos\theta}{r'^2 + r^2}\hat{y}. \tag{37}$$

The normal to the surface is

$$\vec{n} = -\frac{(r'\sin\theta + r\cos\theta)}{\sqrt{r'^2 + r^2}}\hat{x} + \frac{(r'\cos\theta - r\sin\theta)}{\sqrt{r'^2 + r^2}}\hat{y}. \tag{38}$$

Defining $\phi = \arctan r'/r$ we have

$$\alpha^1 = \sigma_z, \tag{39}$$

$$\alpha^2 = \frac{\sin(\phi - \theta)}{\sqrt{r'^2 + r^2}}\sigma_x + \frac{\cos(\phi - \theta)}{\sqrt{r'^2 + r^2}}\sigma_y, \tag{40}$$

and the normal Pauli matrix

$$\beta = \vec{n}\vec{\sigma} = -\cos(\phi - \theta)\sigma_x - \sin(\phi - \theta)\sigma_y. \tag{41}$$

The spin connection is

$$\Gamma_1 = 0, \tag{42}$$

$$\Gamma_2 = -\frac{1}{2}\beta\partial_\theta\beta = \frac{i}{2}(1 - \partial_\theta\phi)\sigma_z. \tag{43}$$

This leaves a final Dirac equation

$$H = -i\left[\sigma_z\partial_z + \frac{\sin(\phi-\theta)\sigma_x + \cos(\phi-\theta)\sigma_y}{\sqrt{r'^2 + r^2}}\left(\partial_\theta + \frac{i}{2}(1 - \partial_\theta\phi)\sigma_z\right)\right]. \tag{44}$$

Now we rotate the Pauli matrices to make them coincide locally with the basis vectors. This is done with the transformation $\tilde{U} = e^{i\sigma_z(\theta-\phi)/2}$, which leads to

$$H = -i\left[\sigma_z\partial_z + \frac{\sigma_y}{\sqrt{r'^2 + r^2}}\partial_\theta\right]. \tag{45}$$

This generalizes the transformation used in Ref. [17] to an arbitrary shape. As this work notes, it is key to realize that $\tilde{U}$ changes the boundary conditions in $\theta$ to antiperiodic because $U(\theta = 2\pi) = -1$. Finally, we make the coordinate change

$$s = \int_0^\theta d\theta' \sqrt{r'^2(\theta') + r^2(\theta')}, \tag{46}$$

$$\frac{\partial s}{\partial \theta} = \sqrt{r'^2(\theta') + r^2(\theta)}, \tag{47}$$

which indeed leads to the Hamiltonian in the flat space form

$$H = -i\left[\sigma_z\partial_z + \sigma_y\partial_s\right]. \tag{48}$$

The coordinate change that brings the equation to flat appearance is an integral equation which in general has no analytic solution except for a few simple cases. But the knowledge of this coordinate change is not needed unless other position dependent terms are to be included in the Hamiltonian.

This derivation appears to show that if the surface has no intrinsic curvature, then the effective Hamiltonian in an appropriate basis has full rotational invariance in the new variable $s$, regardless of the initial cross section. This statement is of course only true to the extent that the linear model is valid. Real wires will only have discrete rotation symmetries, which are apparent when higher order powers or $k$ are included in the continuum Hamiltonian. The cylindrical model is therefore appropriate only up to the energy cutoff given by the coefficient of the quadratic corrections.

In summary, the effective Hamiltonian for a wire of any cross section takes the form of a standard Dirac Hamiltonian in flat space, with antiperiodic boundary conditions. By making the spin basis rotate to follow the basis vectors, we have introduced an extra $\pi$ phase that is often described as the "curvature induced" Berry phase. This is the model used in Ref. [26].

## A.2 Effective Hamiltonian with superconductivity

Given we have presented several different normal Hamiltonians related by local spin rotations, one may wonder whether the formulation of superconductivity still takes its standard form.

In this section we spell out the Bogoliubov-de Gennes formulation explicitly to show that this is the case. In second quantized form, an s-wave pairing term takes the form

$$\mathcal{H}_\Delta = \Delta \psi_\uparrow \psi_\downarrow - \Delta^* \psi_\uparrow^* \psi_\downarrow^* = \frac{1}{2}\big[\Delta \psi^T i\sigma_y \psi - \Delta^* \psi^\dagger i\sigma_y (\psi^\dagger)^T\big], \tag{49}$$

with $\psi = \begin{pmatrix} \psi_\uparrow \\ \psi_\downarrow \end{pmatrix}$ and $\psi^\dagger = (\psi_\uparrow^*, \psi_\downarrow^*)$. This type of term is added to the normal Hamiltonian $\mathcal{H} = \psi^\dagger H \psi$ to model superconductivity. The different normal Hamiltonians in the previous section are related by spin transformations of the form $\psi \to U\psi$ with $U = e^{i\vec{\sigma}\vec{\alpha}/2}$ where $\vec{\alpha}$ are position dependent variables. The Hamiltonian for s-wave pairing in Eq. (49) is not affected by such transformations because

$$\psi^T i\sigma_y \psi \to (U\psi)^T i\sigma_y U\psi = \psi^T e^{i\vec{\sigma}^*\vec{\alpha}/2} i\sigma_y e^{i\vec{\sigma}\vec{\alpha}/2}\psi = \psi^T i\sigma_y e^{-i\vec{\sigma}\vec{\alpha}/2} e^{i\vec{\sigma}\vec{\alpha}/2}\psi = \psi^T i\sigma_y \psi,$$

and the same occurs for the complex conjugate term. This is expected since s-wave pairing forms a spin singlet which is rotationally invariant. The total Hamiltonian $\mathcal{H} + \mathcal{H}_\Delta$ can be rewritten in matrix form in terms of a Nambu spinor $\Psi = \begin{pmatrix} \psi \\ (\psi^\dagger)^T \end{pmatrix}$ as

$$\mathcal{H} + \mathcal{H}_\Delta = \frac{1}{2}\Psi^\dagger \begin{pmatrix} H & -i\sigma_y \Delta^* \\ i\sigma_y \Delta & -H^T \end{pmatrix}\Psi. \tag{50}$$

This is the BdG formulation used in Refs. [23, 24]. This problem can also be described with an alternative basis where $\Psi = \begin{pmatrix} \psi \\ -i\sigma_y(\psi^\dagger)^T \end{pmatrix}$ which means the hole operators are time reversed electron operators. In this basis, the Hamiltonian is

$$\mathcal{H} + \mathcal{H}_\Delta = \frac{1}{2}\Psi^\dagger \begin{pmatrix} H & \Delta^* \\ \Delta & -\sigma_y H^* \sigma_y \end{pmatrix}\Psi. \tag{51}$$

This is the formulation used in this work. Both formulations satisfy particle hole symmetry, $U_C^\dagger H^* U_C = -H$, with $U_C = \tau_x$ for the first and $U = \sigma_y \tau_y$ for the second, where $\tau$ are Pauli matrices in Nambu space.

## A.3 Transfer matrix method

Here we describe the method used to compute the estimate for the gap $\delta$ efficiently. Since we are only interested in states near the Fermi level, we would like to find all values of $k$ (real or complex) for which there is a solution of

$$[H(k) - \mu]\psi_k = 0. \tag{52}$$

If all solutions to this equation are complex, this means there is no propagating state at the Fermi level and the Hamiltonian is gapped. If a real solution is found, then it is gapless. An efficient way to compute $k$ numerically is via the transfer matrix $T$ of the system [50]. The $T$ matrix of a general 1D tight-binding chain with $N$ orbitals per site and nearest neighbor hoppings (with lattice constant $a = 1$) is defined as follows. If the Hamiltonian of the chain is

$$H = u + e^{ik}t + e^{-ik}t^\dagger, \tag{53}$$

where $u$ and $t$ are $N$x$N$ matrices describing all the on-site and nearest neighbour terms, respectively (and it is assumed that $t$ is invertible), the transfer matrix at energy $\epsilon$ is then defined as

$$T = \begin{pmatrix} (t^\dagger)^{-1}(\epsilon - u) & (t^\dagger)^{-1} \\ -t & 0 \end{pmatrix}. \tag{54}$$

An eigenvalue $\lambda$ of $T$ with eigenvector $\psi_\lambda$ satisfies

$$(T - \lambda\mathcal{I})\psi_\lambda = 0 \tag{55}$$

and can be found by solving $\det(T - \lambda\mathcal{I}) = 0$. To see the relation with the eigenstates of $H$, we multiply Eq. (55) by the following matrix

$$M = \begin{pmatrix} t^\dagger & \lambda^{-1}\mathcal{I} \\ 0 & t^\dagger \end{pmatrix} \tag{56}$$

and obtain

$$M(T - \lambda\mathcal{I})\psi_\lambda = \begin{pmatrix} \epsilon - u - \lambda t^\dagger - \lambda^{-1}t & 0 \\ -t^\dagger t & -\lambda t^\dagger \end{pmatrix}\psi_\lambda = 0. \tag{57}$$

This equation implies that

$$\det(t^\dagger)^2 \det(T - \lambda\mathcal{I}) = \det(\epsilon - u - \lambda t^\dagger - \lambda^{-1}t)\det(-\lambda t^\dagger). \tag{58}$$

Since $t^\dagger$ is invertible, if $\lambda$ is an eigenvalue of $T$ we must have

$$\det(\epsilon - u - \lambda t^\dagger - \lambda^{-1}t) = 0, \tag{59}$$

which is the condition for an eigenvalue of $H$ if $\lambda = e^{-ik}$ (with $k$ real or complex). Therefore, the momenta of all propagating and evanescent modes at energy $\epsilon$ can be obtained from the transfer matrix eigenvalues as

$$k = i\log\lambda. \tag{60}$$

Moreover, by writing $\psi_\lambda$ in terms of its block components $\psi_\lambda = (\psi_{1,\lambda}, \psi_{2,\lambda})^T$, the first row of Eq. (57) then implies that

$$(\epsilon - u - \lambda t^\dagger - \lambda^{-1}t)\psi_{1,\lambda} = 0 \tag{61}$$

so that $\psi_{1,\lambda}$ is the eigenvector corresponding to the momentum $k = i\log\lambda$.

To apply this method to the tight binding Hamiltonian defined in the main text in Eq. (18), we Fourier transform the $z$ direction back to real space, where

$$\psi_k^\dagger \cos k_z \psi_k \to \frac{1}{2}(\psi_i^\dagger \psi_{i+1} + \psi_{i+1}^\dagger \psi_i) \tag{62}$$

and

$$\psi_k^\dagger \sin k_z \psi_k \to \frac{i}{2}(\psi_i^\dagger \psi_{i+1} - \psi_{i+1}^\dagger \psi_i), \tag{63}$$

where $i$ denotes the $i$-th site along $z$. These become hopping terms that enter the matrix $t$ in Eq. (53).

To apply this method to a continuum Hamiltonian such as the one in Eq. (3), we need to find a lattice Hamiltonian that reproduces the continuum Hamiltonian in the low energy limit. To do this, we simply replace $\psi_k^\dagger k_z \psi_k \to \psi_k^\dagger \sin k_z \psi_k$ and then Fourier transform back to real space as before. Note this replacement introduces a second low-energy Dirac fermion at $k = \pi$ but this poses no problem for our purposes: eigenstates of the continuum model can be obtained from those of $T$ by selecting those with $Re[k] \leq \Lambda$ with $\Lambda$ a momentum cutoff above which the Dirac model is no longer applicable.



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
