# Peer review of "Conditions for fully gapped topological superconductivity in topological insulator nanowires"

_SciPost Physics, doi:SciPost Phys. 6, 060 (2019)_

## Round 2 · Referee Report · Anonymous · 2018-11-26

Strengths

1. Importance of the subject among the current topics of research in the community
2. Timeliness with respect to experimental developments
3. Appropriate analytical and numerical techniques to approach the problem
4. Clarity of the presentation and of the message
5. Good introduction and description of the results in relationship to previous works
6. Novelty, in particular important differences from the conclusions of similar works addressing this subject in the past

Weaknesses

1. The physical significance of the terms in Eq. 20 in not given, the coupling lambda's for example are not introduced, there is only a reference to a model in [23] which in my opinion is not enough.
2. The origin of the discrepancy with Ref. [24] is very vaguely explained, especially since the differences from Ref.[24] seems to be one of the central and most important points of the paper. I believe a more thorough analysis is justified, especially in what concerns "the importance of choosing a suitable vector potential" and the "inhomogeneous supercurrent profile"; more generally the manuscript would benefit from a discussion of the relationship between the supercurrent profiles arising, their conservation and physical-ness in relationship also with the boundary conditions, and their dependence on choosing a gauge in finite-size discretized models.

Report

This is a careful, clear, detailed and actual work concerning the formation of Majorana states in finite-width wires threaded by a magnetic flux. In particular the authors explore the topological phase diagram and the modification of the gap protecting the eventual topological states both in the presence and absence of a vortex in the superconducting providing the proximity effect necessary for the formation of Majorana states. Interestingly enough they find that when there is no vortex in the SC, such states cannot form in general due to a gaplessness of the system, unless the symmetry is either fully broken by geometry, disorder or inhomogeneities, but even so the induced gap is much smaller than when a vortex is present in the SC. The present results differ from a similar previous analysis which suggested that even in the absence of a vortex the Majorana states are protected by a topological gap.

I would recommend the publication of this manuscript in Scipost if the requested changes are made and the questions raised are answered, in particular if a careful analysis of the discrepancy with Ref. [24] and an analysis of the supercurrent profiles and of the gauge choices are performed.

Requested changes

1. Clarifications of the terms in Eq. 20.
2. Introduction of a thorough analysis of the significance of the choice of gauge, of the super current profiles, and of the origin of the discrepancy with the results of Ref. [24].

  • validity: good
  • significance: top
  • originality: high
  • clarity: top
  • formatting: perfect
  • grammar: perfect

Author Fernando de Juan on 2019-04-09 (in reply to Report 1 on 2018-11-26)

We would like to thank the referee for their detailed report on our manuscript. We have carefully considered all the issues raised and modified the text to address them.

- "The physical significance of the terms in Eq. 20 in not given, the coupling lambda's for example are not introduced."

We have now introduced explicitly the significance of every parameter in the lattice model after Eq. 20. Note also we have relabeled the orbital degree of freedom from $\sigma$ to $\rho$ in the lattice model, to avoid confusion with the continuum model section where $\sigma$ was the effective spin degree of freedom of the surface states.

- "The origin of the discrepancy with Ref. [24] is very vaguely explained, especially since the differences from Ref. [24] seems to be one of the central and most important points of the paper. I believe a more thorough analysis is justified, especially in what concerns "the importance of choosing a suitable vector potential" and the "inhomogeneous supercurrent profile"; more generally the manuscript would benefit from a discussion of the relationship between the supercurrent profiles arising, their conservation and physical-ness in relationship also with the boundary conditions, and their dependence on choosing a gauge in finite-size discretized models."

We should first emphasize that we have not found a definite reason for the discrepancy with Ref. [24]. We can only provide the evidence that continuum and lattice models do match in our case when symmetries are implemented properly. Then we provide the example of the symmetry-breaking supercurrent only as one plausible reason for the discrepancy. We do agree with the referee that the discussion about this point in the main text was not very detailed, so we have extended it significantly in Sec. 3.1. We believe the issue is much better explained in the current version.

---

## Round 2 · Referee Report · Anonymous · 2018-12-3

Strengths

1- It deals with a very timely and interesting problem: the topology of spin-orbit-coupled nanowires embedded in a SC shell and threaded by a magnetic flux

2- It includes some detailed derivations that could be useful for students and researchers entering this subfield

3- The quality of the presentation is decent, although somewhat verbose

4- Quite a few aspects of the problem are covered, including the role of lattice symmetries and supercurrent inhomogeneities

Weaknesses

1- It has almost a 100% overlap with Ref. 48

2- It does not clarify whether any of the conclusions is different in a TI nanowire, respect the semiconducting nanowires of Ref. 48. Particularly, the most problematic result (gapless spectrum in clean systems) seems to be the same in both.

Report

The study presented in this work has a very high overlap with Ref. 48, posted in the arXiv more than a month before this preprint. To accept this work for publication, it should highlight the differences with Ref 48 very clearly. In some aspects Ref. 48 does a considerably more thorough job in analysing the complicated physics of full-shell semiconducting nanowires, particularly as it probably had the benefit of knowing about the experimental results beforehand. This work should compensate for this in some way, or risk being ignored. Given my particular interest in seeing the SciPost initiative succeed (i.e. become a relatively high-impact journal), I would insist that the authors make this paper as competitive as possible with Ref. 48 before I can recommend publication. Essentially, it needs to be shown that the TI approach is different than the semiconducting nanowire approach, and is better in some significant sense. In more detail, I would suggest that the following issues are addressed.

It seems that the model employed here is essentially equivalent to that of Ref 48, except in the lack of a p^2 term in the Hamiltonian. If this is not so, please point out the differences very clearly. If that is indeed the only difference, does it lead different conclusions at all? In Ref. 48, it was already shown, for example, that the h/2e flux can only gap one mode, that with zero total angular mom,entum, while any higher angular momenta that are occupied will remain gapless. This seems also to be the case here, as eta=1/2 can only Cooper-pair one subband at zero energy. The gapless results in Sec. 3.1 seems to confirm this, unless rotation symmetry is broken (this possibility is also extensively discussed in Ref. 48). Unfortunately, the real reason for the gapless spectrum (gapless |l|>=1 modes) is not clearly given in this manuscript. As the occupation of different subbands of higher l is not readily tuneable in full shell nanowires due to electrostatic screening by the SC, and as rotation symmetry breaking is likely a small uncontrolled perturbation, it is not clear that the claim that a h/2e vortex can in general gap the system is actually justified.

Constraining the model of the TI nanowire to just the surface is very artificial. A thin nanowire will probably have considerable leakage of the surface wavefunction into the core. This scenario is nicely included in the lattice calculation of section 3, which shows that leakage doesn't seem to change the conclusions. It would be important to clarify under which conditions the topological transitions survive this leakage, and why, as it is not at all clear from the analytic description of the problem.

Minor issues:

1 - There are two definitions of the flux quantum, the Dirac flux quantum h/e, relevant to the Aharonov-Bohm effect, and the superconducting flux quantum h/2e, relevant to superconducting vortices. The text should clarify that they are using the former definition when talking about "half a flux quantum".

2 - The manuscript says: "In a more thin film geometry, one may rather expect a roughly homogeneous order parameter which can be approximated by a constant ∆(x,θ)=∆, so nv=0". I don't fully understand what it is meant by this. The statement seems counterintuitive to me. A thin SC shell should more easily relax to the fluxoid of minimum energy, which will not be nv=0 as soon as the flux is greater than h/4e.

3 - The derivation of the continuum Hamiltonian in Sec. 5 seems much more complicated than is really necessary, and involves a number of spin rotations and change of variables. For comparison, the derivation in Ref. 48 is much simpler. Please try to present a more compact derivation of the Hamiltonian if possible.

4 - The topological invariant is defined in term of Pfaffians at high-symmetry points. Kitaev introduced this in a single-mode model. Is it justified in a generic multimode context with spin-orbit mode mixing? Does the topological gap inversion always take place at high-symmetry points in this case? Please give references.

Requested changes

1- Extend the paper with a comprehensive comparison to Ref. 48, highlighting relevant differences. If there are none that are relevant, I would not recommend publication.

2- Explain why bulk leakage doesn't change the results of the hollow core model.

3- Explain the origin of the gapless spectrum more clearly

4- Adresss "minor issues" 1 to 4 in the report.

  • validity: top
  • significance: top
  • originality: poor
  • clarity: ok
  • formatting: good
  • grammar: excellent

Author Fernando de Juan on 2019-04-09 (in reply to Report 2 on 2018-12-03)

We thank the reviewer for their report on our manuscript. We share the reviewer's enthusiastic support of the SciPost journal, and are happy to provide detailed answers to the reviewer's concerns. Since most of the issues raised stem form a comparison with Ref. 48, and the claim that our work has "almost 100 % overlap" with it, we will first summarize why we believe this is not a valid criticism and provide a detailed discussion afterwards.

Ref. 48 studies proximitized Rashba split semiconductor nanowires, originally proposed in Refs. 6-7, while our work studies proximitized TI nanowires, proposed in Ref. 23. If the superconductor used to induce proximity effect is a full shell surrounding the wire, the possible winding of the order parameter phase around the wire (a vortex henceforth) becomes relevant in both cases. For semiconductor nanowires, Ref 48 uses a model with such winding which can be mapped to the model in Refs. 6-7. For TI nanowires, the influence of this winding and the relevance of angular momentum conservation was already discussed in our previous paper Ref. 26 in 2014, which is cited several times in our manuscript when these issues are discussed. The claim that our work has "almost 100 % overlap" with Ref. 48 boils down to the fact that the low-energy continuum model used in both works is in appearace similar, and that Ref. 48 makes predictions about gaplessness due to angular momentum conservation. However, the model for TI nanowires is not in any way based on Ref. 48, but rather it has been proposed before. The physical differences with semiconductor nanowires are well known, in particular the key fact that for half a flux quantum one gets an odd number of modes and therefore a topological superconductor for any chemical potential within the bulk gap (Ref. 23). The relevance of angular momentum conservation and the presence of the vortex was already discussed in Ref. 24 and in our work in Ref. 26, which is not discussed or cited in Ref. 48. Our paper is an independent piece of work, which discusses a different system with different features, building on several previous works that are acknowledged throughout the manuscript. Therefore we do not believe that ”almost 100 % overlap” with Ref. 48 is a fair characterization of our work. Put simply, one cannot infer the physics we account for from the components of Ref 48. Nevertheless, after considering the referee's comments we agree that a more detailed comparison with Ref. 48 in the text will benefit the reader, and we have included it in the discussion section.

A more detailed answer to the reviewer now follows. To provide more context to our answers, we first present a chronological review of the relevant literature. The proposal of topological superconductivity in topological insulator nanowires was put forward by Cook and Franz in Ref. 23 of our manuscript, in 2011. A subsequent study (Ref. 24) in 2012 reported a puzzling result regarding the influence of a vortex in the topological nature of the system. In 2014, we, the authors, published a paper (Ref. 26) dealing with transport in this system, using a very similar formalism to the one used in our work now submitted to SciPost. In particular, after Eq. (4) in Ref. 26, we discuss extensively the role of angular momentum conservation in the rotationally symmetric model, stating that the lowest energy electron and hole branches cannot mix with each other because they do not have the same angular momentum, unless a vortex is present to compensate the mismatch. The implications of this statement, which eventually gave rise to our current work, have been presented by the three authors in conferences ever since 2014, starting with the 2015 March Meeting (http://adsabs.harvard.edu/abs/2015APS..MARG12009D). Also in 2014, Ref. 27 reported more realistic numerical simulations of TI nanowires in different geometries which also displayed some unexpected gapless regions. Finally, in Ref 42 (2018), a model for full shell Rashba-split semiconducting wires was described. This coupled-chain model is different from that in Ref. 48, but already discussed that with discrete rotation symmetry, gapless regions are generic, and that these become gapped upon breaking of the symmetry, with small gaps determined in magnitude by the perturbation.

- "It seems that the model employed here is essentially equivalent to that of Ref 48, except in the lack of a p$^2$ term in the Hamiltonian. If this is not so, please point out the differences very clearly. If that is indeed the only difference, does it lead different conclusions at all?"

The reviewer is correct that the effective model in Eq. (5) of Ref. 48 can be formally obtained from the model for proximitized TI surface states by the addition of the $p^2$ term. Indeed, Ref. 48 shows that this model maps to the original model for Rashba-split nanowires (Refs. 6-7) in the hollow-core limit. The differences between the model in Refs 6-7 and the TI nanowire model in Ref. 23 are well known. The Rashba band structure of semiconductor nanowires only has an odd number of modes for chemical potentials within the Zeeman gap, which is typically very small (order meV), while the bands of TI nanowires have an odd number of modes \textit{for any value of the chemical potential within the bulk gap}, 300 meV in Bi2Se3. This feature must remain in any model for TI nanowires at any order in momentum, as it is a topological property. Since having an odd number of modes is the essential requirement to produce a topological superconductor, TI nanowires have a critical advantage compare to semiconductor ones.

Moreover, the model in Ref. 48 has an effective time-reversal symmetry when $\eta = n_v/2$ in our notation, or $\phi =0$ in the notation of Ref. 48. Our model also has this symmetry. However, since the model in Ref. 48 derives from a bulk 1D system, the topological index must be zero in the presence of time-reversal symmetry, which is a general constraint on class D superconductors. The topological regions in Ref. 48 always appear when $\eta \neq n_v/2$ (or finite $\phi$ in their notation), and require tuning of the chemical potential. Our work shows that in the presence of bulk inversion symmetry, there is always a gapped topological state for $\eta = n_v/2$ for an arbitrary value of the chemical potential. This conclusion is indeed confirmed with a lattice model, and represents a key difference between Ref. 48 and our work.

The effect of inversion and time-reversal symmetries was not sufficiently explained in the previous version of the manuscript. We have enlarged this discussion in the main text, in particular rewriting section 2.1 almost completely, and have brought the appendix describing time-reversal to this section to avoid repetition. We have also included a paragraph in the discussion section that comments on the differences with Ref. 48.

- "In Ref. 48, it was already shown, for example, that the h/2e flux can only gap one mode, that with zero total angular mom,entum, while any higher angular momenta that are occupied will remain gapless. This seems also to be the case here, as eta=1/2 can only Cooper-pair one subband at zero energy. The gapless results in Sec. 3.1 seems to confirm this, unless rotation symmetry is broken (this possibility is also extensively discussed in Ref. 48). Unfortunately, the real reason for the gapless spectrum (gapless $|l|>=1$ modes) is not clearly given in this manuscript. As the occupation of different subbands of higher l is not readily tuneable in full shell nanowires due to electrostatic screening by the SC, and as rotation symmetry breaking is likely a small uncontrolled perturbation, it is not clear that the claim that a h/2e vortex can in general gap the system is actually justified."

We must first emphasize that our results are very different to what the referee is describing. Both Fig. 3 from the continuum model and Fig. 5a for the tight binding model (with fourfold rotation symmetry) show a region around $\eta = 0.5$ which is gapped for an arbitrary value of the chemical potential, and thus an arbitrary occupation of the higher angular momentum modes. Indeed, in the tight binding simulation, a gapped, topological state is obtained for any chemical potential within the bulk gap. The number of diamonds in the vertical direction corresponds with the number of modes occupied. This requires no disorder at all. Therefore, our results were not shown in Ref. 48, where the occupation of any mode of higher angular momentum renders the spectrum gapless. This is all in the case of winding $n_v=1$ of the order parameter, i.e. one vortex. The reason for a gapless spectrum in the case without a vortex, $n_v=0$, was already explained in our 2014 work of Ref. 26 for the lowest occupied mode.

We do agree with the reviewer that the fate of the higher angular momentum modes is however not discussed with sufficient clarity in our manuscript. Previously only the lowest mode was discussed in the continuum model section, and a few cases relevant to discrete rotation axes were discussed in the tight binding model. In light of this discussion, we have significantly extended those sections to explain what happens. We thank the referee for making this point.

- Constraining the model of the TI nanowire to just the surface is very artificial. A thin nanowire will probably have considerable leakage of the surface wavefunction into the core. This scenario is nicely included in the lattice calculation of section 3, which shows that leakage doesn't seem to change the conclusions. It would be important to clarify under which conditions the topological transitions survive this leakage, and why, as it is not at all clear from the analytic description of the problem.

This problem is very well characterized both in theory and in experiment in TI nanowires, and it is indeed the case that the surface model provides a realistic account of actual physical systems. Essentially, the surface states are exponentially localized at the surface with a decay length of $v_F/E_g$ with $v_F = 330 \rm{meV nm}$ the Fermi velocity and $E_g = 300 \rm{meV}$ the bulk gap for Bi2Se3. The decay length is of the order of $\rm{nm}$ and in actual experiments the surface state is well localized and bulk leakage is not a problem. In addition, Refs 23-24 showed at the level of a tight-binding model that the surface state dispersion of a finite wire of the same dimensions as we use follows the effective model quite well. Our own work shows that a triangular wire, for example, is still described by the effective model except for a shift of the value of the flux that realizes an odd number of modes. Finally, this good match persists even at the level of ab-initio calculations, such as those in Ref. 41. Experimentally, wires which are insulating enough in the bulk that they display well developed surface transport with the expected Aharonov-Bohm oscillations (Refs. 32 and 34) and even surface Andreev reflection (Refs. 39-40) have been already realized. We have added a sentence before the effective model is discussed where we summarize when the effective model should apply.

- 1) "There are two definitions of the flux quantum, the Dirac flux quantum h/e, relevant to the Aharonov-Bohm effect, and the superconducting flux quantum h/2e, relevant to superconducting vortices. The text should clarify that they are using the former definition when talking about "half a flux quantum" "

We agree with the reviewer that this point could have been explained explicitly, and we do so now in the introduction where the flux quantum is first mentioned.

- 2) "The manuscript says: "In a more thin film geometry, one may rather expect a roughly homogeneous order parameter which can be approximated by a constant $\Delta(x,\theta)=\Delta$, so $nv=0$". I don't fully understand what it is meant by this. The statement seems counterintuitive to me. A thin SC shell should more easily relax to the fluxoid of minimum energy, which will not be nv=0 as soon as the flux is greater than h/4e.

This sentence was aimed to be a restatement of the explanation given in the introduction, which refers to the two situations in Fig. 1. Fig 1 is also mentioned further up before the sentence the reviewer refers to. What is meant is that the superconductor is flat thin film, and the wire lies on top of it, so that the wire is contacted only partially. This is a very different situation from a full shell wire. We have rewritten the text so that no potential confusion arises.

- 3) "The derivation of the continuum Hamiltonian in Sec. 5 seems much more complicated than is really necessary, and involves a number of spin rotations and change of variables. For comparison, the derivation in Ref. 48 is much simpler. Please try to present a more compact derivation of the Hamiltonian if possible."

The derivation of the TI wire surface state Hamiltonian has already been provided in a number of references [17,18,23,24] with different approaches. As we state in the opening paragraph of section 5, which is an appendix, the emphasis is on unifying those previous approaches. This naturally provides more information than one concise derivation from a particular approach. Because this is an appendix, we have also chosen to present a more general derivation, allowing for an arbitrary cross section of the wire in which the surface Dirac fermion moves. We believe this to be a useful information for various potential extensions of our work, and it would not serve any real purpose to simplify this derivation by making it less general.

- 4) "The topological invariant is defined in term of Pfaffians at high-symmetry points. Kitaev introduced this in a single-mode model. Is it justified in a generic multimode context with spin-orbit mode mixing? Does the topological gap inversion always take place at high-symmetry points in this case? Please give references."

The topological invariant defined by Kitaev in his original paper, Ref. [9], applies to a generic system with any number of bands, see Eq. (18). The general invariant is given in Eq. (26) in terms of the Pfaffians at momenta 0 and $\pi$, which is our Eq. (8). Reference to Kitav's paper was naturally already given at the point that the invariant was introduced and no further references are needed.

Requested changes:

1- "Extend the paper with a comprehensive comparison to Ref. 48, highlighting relevant differences. If there are none that are relevant, I would not recommend publication."

In light of our detailed explanations in this response, we believe the literature already has an extensive discussion of the physical differences between topological insulator nanowires and semiconducting nanowires, and that we have cited all relevant works explaining this in our paper. We believe that this is the best resource to understand the differences between our work and Ref. 48. Nevertheless, we agree with the reviewer that an explicit comparison of both works would benefit the reader, so we have included an extra paragraph at the end of the discussion section discussing the most important difference between these two systems in the full-shell geometry, which is that the TI nanowire system has a topological, gapped region around $\eta = n_v/2$ which extends to arbitrary values of the chemical potential, while this is not the case for semiconducting nanowires.

2- "Explain why bulk leakage doesn't change the results of the hollow core model."

See above.

3- "Explain the origin of the gapless spectrum more clearly"

We have updated the text to clarify this point.

4- "Adresss "minor issues" 1 to 4 in the report."

See above.

Anonymous on 2019-05-06 (in reply to Fernando de Juan on 2019-04-09)

The authors have provided an excellent reply to the referee report, which I now realise contained numerous mistakes and misunderstandings. The historical perspective of the problem given in the reply is very useful. The detailed comparison to Ref 47's theory is enlightening, and clearly lays out important and relevant differences from that work, and at the same time adds substantial appeal to the TI approach explored in this submission. I fully recommend publication.

---

## Round 3 · Referee Report · Anonymous (Referee 2) · 2019-5-6

Report

The authors have provided an excellent reply to the referee report, which I now realise contained numerous mistakes and misunderstandings. The historical perspective of the problem given in the reply is very useful. The detailed comparison to Ref 47's theory is enlightening, and clearly lays out important and relevant differences from that work, and at the same time adds substantial appeal to the TI approach explored in this submission. I fully recommend publication.

---

## Round 3 · Referee Report · Anonymous (Referee 1) · 2019-5-6

Report

In my opinion the authors have answered satisfactorily to the recommendations in the previous reports, so the manuscript can be published in Scipost in the present form.

---

## Round 3 · Author Response

We have considered all issues and criticisms raised by the reviewers, and we have modified our manuscript to address them. Please the referee reports for detailed answered.

---

## Round 3 · List of Changes

• We have stated explicitly that we consider the Aharonov-Bohm flux quantum h/e, and that half a flux quantum refers to h/2e.
  • At the beginning of section 2 we have explained why the surface model is appropriate and under what conditions.
  • We have clarified what we meant by the "thin film limit" of the bulk superconductor in the text.
  • We have done a major rewriting of section 2.1 simplifying the equations and addressing the general case of arbitrary $\eta$, $\l$ and $n_v$. We have included the operators for time-reversal and inversion symmetries.
  • We have also rewritten section 2.2 to discuss the implications of time-reversal symmetry in bulk vs surface models. We have merged into this discussion what used to be appendix 5.3.
  • We have emphasized throughout the text that when $\eta = n_v/2$ we get a topological state for an arbitrary value of the chemical potential.
  • For the lattice model, we have defined the operators of inversion and mirror symmetries.
  • We have introduced the significance of every parameter in the lattice model after Eq. 20. Note also we have relabeled the orbital degree of freedom from $\sigma$ to $\rho$, and we have made consistent the notation for the transformation to the Majorana basis.
  • We have significantly extended the discussion on the discrepancy with Ref. 24 in Sec. 3.1, elaborating on the choice of gauge and inhomogeneous supercurrents.
  • We have significantly extended the discussion of the results of the lattice model calculations with difference discrete rotation symmetries, and explained the origin of the gapped and gapless regions.
  • In the discussion section, we have included a paragraph about possible mechanisms that can break the effective time-reversal symmetry at $\eta = n_v/2$.
  • We have removed that added note about Refs. 47 and 48, and instead included another paragraph in the discussion section commenting on the main differences between our work and Ref. 48. (note in the current manuscript Ref numbers 47 and 48 are interchanged compared to the original manuscript). In our response to referees we have used the numbering corresponding to the original manuscript.
  • We have also fixed several minor typos through the text.

---

## Editorial Decision

published